# Quality and Nutritional Value of 'Chopin' and Clone 'JB' in Relation to Popular Apples Growing in Poland

**Andrii Kistechok \*** , **Dariusz Wrona** and **Tomasz Krupa**

Department of Pomology and Horticulture Economics, Institute of Horticultural Sciences,
Warsaw University of Life Sciences (SGGW-WULS), 159C Nowoursynowska Street, 02-787 Warsaw, Poland
\* Correspondence: andrii_kistechok@sggw.edu.pl; Tel.: +48-22-593-20-81

**Abstract:** The aim of the study was to describe the physicochemical properties, with particular emphasis on nutritional value, of apples 'Chopin' and clone 'JB'. The new cultivars were compared with the cultivars commonly cultivated in Polish orchards, namely 'Gala Brookfield', 'Šampion', 'Ligol' and 'Idared'. The study focuses on the evaluation of physicochemical characteristic (firmness, soluble solids content and titratable acidity) and the content of monosaccharides, sucrose and organic acids based on HPLC with an RI detector. In addition, the nutritional value of apples were described separately for the flesh and peel of the fruit, focusing on the assessment of the antioxidant activity and the content of total polyphenols, phenolic acids, flavonols using the HPLC technique. 'Chopin' and 'JB' clone apples are characterized by very high acidity, over 1%, which is related to the high content of malic acid. The red flesh 'JB' clone is characterized by a high content of bioactive compounds in both the peel and flesh of apples. High temperatures and a lack of precipitation contribute to a higher polyphenol content in apples, which proves that apart from the genetic features of the cultivars, the climatic conditions also determine the nutritional value of the fruit.

**Keywords:** apple; firmness; SSC; monosaccharides; titratable acidity; acids; total polyphenols; antioxidant activity

## 1. Introduction

Poland, Spain and Italy are the largest fruit producers in Europe. Poland is especially prominent in the production of apples, cherries, raspberries, currants and gooseberries [1,2]. Comparing apple production in Poland and other European Union countries, we see that the same apple cultivars are very popular in different countries. In the European Union, 'Golden Delicious', 'Idared', 'Jonagold' and its mutants, and 'Gala' and its mutants are the most widespread [2]. On the other hand, the Central Statistical Office reports that in Poland [3], the most widespread include 'Idared' (22% of the total apple area in Poland), 'Jonagold' (17%) and 'Gala' (6%). Apple cultivars are characterized by different consumption and sensory attributes, which has been proven in numerous studies [4,5]. According to WHO recommendations, the daily intake of fruit and vegetables should be a minimum of 400 g [6]. Fruit and vegetables are an important component of diets and appropriate dietary programmes can reduce the risk of lifestyle diseases [7]. It is generally recommended that vegetable intake should be higher than fruit intake [8]. However, in a properly designed nutrition program, fruit cannot be omitted, although the benefits of fruit consumption are related to the characteristics of specific fruits. In the National Health and Nutrition Examination Survey (NHANES) 2003–2010, it was shown that the inclusion of any apple in children's diets contributed to a reduction in children's weight as well as influencing higher overall fruit intake [9]. Furthermore, according to Hodgson et al. [10], daily apple consumption reduces the risk of mortality among elderly women caused by various causes, including cancer. Similarly, a review by Gayer et al. [11] showed that apple consumption significantly reduces body weight, significantly reduces the risk of

cerebrovascular disease, cardiovascular death, type 2 diabetes and mortality, reducing the overall risk of cardiovascular disease.

By analysing the ingredients of various diets, it can be concluded that fruit is an important element in many weight-loss diets [12,13], and that its dietary benefits are due to its nutritional properties. Apples contain numerous compounds with antioxidant activity, e.g., ascorbic acid and from the polyphenol group in the form of simple phenols, benzoic acids, phenyl propanoids and flavonoids [14]. Flavonoids and phenolic acids are the predominant polyphenol groups in apples [15,16], but their content depends on the part of the fruit analysed. Significant levels of flavonols are found in the skin of apples [15], while numerous phenolic acids can be found in the flesh [17]. The content of biologically active compounds in apples is determined by many factors: cultivar [18,19], degree of ripeness [20] or fruit fragment [21,22] and geographic area [23,24].

A priority indicator of apple consumer quality is the attractive appearance of the fruit and, in the case of red apples, skin colour is very important [25]. However, the texture of apples is a second extremely important sensory attribute influencing consumer liking, which determines the final acceptance and habituation of the consumer to a particular apple cultivar [5]. Texture, in combination with taste and appearance of apples, resulting from consumer tastes, becomes one of the main priorities in breeding and varietal selection [26,27]. Important components of apple texture are considered to be crispness, firmness and juiciness but also skin firmness [5]. In addition, high firmness combined with juiciness are very positively perceived in consumer evaluation [28].

The aim of the study was to describe the physicochemical characteristics and nutritional value of the fruit of two Polish apple cultivars. In the study, individual fruit characteristics of the 'Chopin' cultivar and the 'JB' clone were compared with commonly grown apple cultivars in Poland. The evaluation focused on the characteristics that distinguish the two cultivars in the content of biologically active compounds and, in addition, on the physicochemical characteristics (firmness, sugar content and acidity), determinants in the positive perception of apples.

## 2. Materials and Methods

The fruit of the new cultivar 'Chopin', clone 'JB' and most popular cultivars, such as 'Gala Brookfield', 'Idared', 'Šampion' and 'Ligol', came from the Experimental Field of the Institute of Horticultural Sciences 'Wilanów', located in central Poland (52.259° N, 21.020° E). Apples were collected from 8-year-old trees growing on M.9 rootstock at a spacing of 1.0 m × 3.5 m. Fruits were harvested from a random selection of 10 trees per row, respectively, for each cultivar. The 'Chopin' cultivar obtained by Prof. E. Pitera is a cultivar promoted for cultivation in Poland, previously described in publications [29]. In 2016, it was entered into the cultivar register of the Research Centre For Cultivar Testing (COBORU) under the number S616 and is a protected by right (PBR) cultivar. It is a single-color cultivar with green skin, without blush. Clone 'JB' is at the preliminary stage of registration and the author is Prof. A. Przybyla [30]. A unique feature is the red flesh of the 'JB' clone. Both cultivars are resistant to apple scab.

The research was conducted in 2019–2021. The weather data are shown in Figures S1–S3. To illustrate climatic conditions, the climatogram method developed by Walter [31] was used, which involves graphically depicting average monthly temperatures and monthly precipitation totals in a ratio of 1 °C:4.5 mm of precipitation. The year 2019 was warm and dry. Precipitation deficits occurred from June to November. In 2020, heavy precipitation occurred in June (more than 140 mm) and in September and October. In 2021, July and August were very wet (precipitation of about 140 mm), while a dry period occurred in September and October. Average temperatures in 2020 and 2021 were around 20 °C in July and August but both years were cooler than 2019.

Apples were characterized in terms of physicochemical quality by assessing firmness, soluble solids content (SSC) and titratable acidity (TA). In addition, the nutritional properties of apples were presented, focusing on the evaluation of antioxidant activity (AA) and

the content of total polyphenols (TPC), phenolic acids, flavonols, sugars and organic acids. The analyses of antioxidant properties were assessed separately in fruit peel and flesh. The fruit samples for assessing nutritional value, separately taken from peel and flesh, were immediately frozen in liquid nitrogen and stored ($-80$ °C) after collection. The peel of apples was taken from opposite sides of the fruit, vertically (from top to bottom). All peel fragments of 10 fruits from the replicate (mixed test) were ground in an IKA A11 grinder (IKA Werke, Staufen, Denmark) in liquid nitrogen. Fruit flesh was sampled similarly, cutting 2 opposite segments from 10 apples in replicate. These analyses were carried out in 4 replicates for each cultivar.

Measurements included the evaluation of the ethylene content of the seed chambers (IEC) and the Streif index and starch index to determine the optimum harvest date. The ethylene content in the seed chambers ($\mu$L/L) was assessed according to a commonly used method [5]. It was measured in the core space of the apples using a 1 mL syringe for air sampling. From each apple, 1 mL of air was taken and ethylene content was assessed using a gas chromatograph (HP 5890, Hewlett Packard, Palo Alto, CA, USA). The starch index (SI) was estimated according to Tomala et al. [32]. The measurement was based on the evaluation of the reaction of starch with Lugol's solution and was visually assessed on a 10-point scale. The Straif index was evaluated based on three components, i.e., firmness, soluble solids content and starch index, according to the formula:

$$\text{Index Streif} = \frac{\text{firmness}}{\text{soluble colids content} \times \text{starch index}}$$

Firmness was analysed after removing the skin from the apple and was measured on two opposite sides of the fruit for each apple. The measurement was performed with an Instron 5542 penetrometer (Instron, Norwood, MA, USA), using a stainless steel plunger tip (diameter: 11 mm; head speed: 240 mm/min) [32,33]. Results are presented in Newtons (N). The evaluation of soluble solids content was performed after squeezing juice from 10 apples. A digital refractometer PR-32 (Atago Co., Ltd., Tokyo, Japan) was used for the measurement [32,33]. The results of the evaluation of soluble solids content were expressed in °Brix. Titritable acidity was assessed in juice diluted with distilled water in a ratio of 1:10. The measurement was carried out using a TitroLine 5000 automatic titrator (Xylem Analytics Germany GmbH, Weilheim, Germany) by titrating the resulting mixture with NaOH (0.1 M) to a pH value of 8.1 [32,33]. The results of the titratable acidity assessment were expressed in % after conversion to malic acid equivalent.

Total polyphenol content was estimated using the spectrophotometric method [34] with the Folin–Ciocalteu reagent. The absorbance of the solution was measured using a Marcel 330S PRO spectrophotometer (Marcel, Zielonka, Poland). The measurement was performed at $\lambda = 700$ nm and the results were converted from the curve to gallic acid (mg$\cdot$100 g$^{-1}$ FW).

The quantitative and qualitative analysis of phenolic compounds (phenolic acids and flavonols) was performed using the HPLC technique described in our earlier study [35]. For the analysis, we used a Perkin-Elmer 200 series HPLC kit with a DAD (Diode Array Detector) using a LiChroCART 125-3 column (Merck KGaA, Darmstadt, Germany). The following analysis parameters were used: flow rate of 1 mL/min and oven temperature of 22 °C. The mobile phase consisted of a mixture of water (A), 20% formic acid (B) and acetonitrile (C) at different concentration gradients. The content of individual compounds (total of two groups: phenolic acids and flavonols) is given in (mg$\cdot$100 g$^{-1}$ FW).

Antioxidant activity was evaluated according to the method of Saint Criq de Gaulejac et al. [36] using the synthetic radical DPPH (1,1-diphenyl-2-picrylhydrazine, Sigma-Aldrich, Poznań, Poland). Results were expressed in mg per g FW of ascorbic acid (AAE). (mg AAE$\cdot$100 g$^{-1}$ FW).

The content of sucrose and monosaccharides and organic acids (malic, citric, tartaric) were determined in 2021 year by HPLC-RI, as described previously by Zielinski et al. [37], and expressed as grams of sugar content or organic acid per 100 g fresh weight (FW).

*Statistical Analysis*

The results were analysed statistically in Statistica 13.3 (StatSoft Polska, Krakow, Poland), using the two-way analysis of variance. Tukey's test was used for the evaluation of the significance of differences between the means, accepting the significance level as 5%.

## 3. Results

The maturity stage of the fruit is shown in Table 1. Three indices were used to determine the optimal harvest time: internal ethylene content, starch index and Streif index. The tests used indicate that the apples had very similar maturity at harvest in all years of the study.

**Table 1.** Characteristic of the maturity stage of apples assessed directly after harvest.

| Cultivars | Years | | |
|---|---|---|---|
| | **2019** | **2020** | **2021** |
| | Internal ethylene content (µL/L) | | |
| 'Gala Brookfield' | $1.04 \pm 0.50$ | $0.74 \pm 0.58$ | $1.41 \pm 0.05$ |
| 'Šampion' | $0.52 \pm 0.60$ | $1.16 \pm 0.01$ | $1.18 \pm 0.04$ |
| 'Ligol' | $1.75 \pm 0.50$ | $1.30 \pm 0.08$ | $1.13 \pm 0.02$ |
| clone 'JB' | $0.79 \pm 0.30$ | $0.82 \pm 0.10$ | $0.70 \pm 0.18$ |
| 'Chopin' | $1.40 \pm 0.90$ | $1.32 \pm 0.25$ | $1.40 \pm 0.07$ |
| 'Idared' | $1.41 \pm 0.10$ | $1.27 \pm 0.04$ | $1.42 \pm 0.06$ |
| | Streif index $(-)$ | | |
| 'Gala Brookfield' | $0.11 \pm 0.008$ | $0.10 \pm 0.010$ | $0.11 \pm 0.011$ |
| 'Šampion' | $0.07 \pm 0.010$ | $0.07 \pm 0.004$ | $0.08 \pm 0.007$ |
| 'Ligol' | $0.07 \pm 0.004$ | $0.08 \pm 0.007$ | $0.07 \pm 0.005$ |
| clone 'JB' | $0.07 \pm 0.006$ | $0.07 \pm 0.005$ | $0.09 \pm 0.008$ |
| 'Chopin' | $0.09 \pm 0.007$ | $0.09 \pm 0.009$ | $0.09 \pm 0.007$ |
| 'Idared' | $0.07 \pm 0.003$ | $0.09 \pm 0.008$ | $0.08 \pm 0.007$ |
| | Starch index $(-)$ | | |
| 'Gala Brookfield' | $6.2 \pm 0.5$ | $7.4 \pm 0.2$ | $6.7 \pm 0.4$ |
| 'Šampion' | $6.4 \pm 0.4$ | $6.6 \pm 0.6$ | $6.1 \pm 0.2$ |
| 'Ligol' | $6.4 \pm 0.3$ | $7.1 \pm 0.9$ | $6.6 \pm 0.5$ |
| clone 'JB' | $6.5 \pm 0.2$ | $7.7 \pm 0.1$ | $6.2 \pm 0.5$ |
| 'Chopin' | $6.2 \pm 0.1$ | $7.4 \pm 0.1$ | $6.6 \pm 0.3$ |
| 'Idared' | $6.9 \pm 0.1$ | $7.1 \pm 0.2$ | $7.6 \pm 0.1$ |

Data are presented as mean ± standard deviation.

One of the most important quality indicators influencing consumer acceptance and decisions is fruit firmness. Among the cultivars studied, priority is given to 'Gala Brookfield', characterized by the highest firmness in all years of the study (Table 2). 'Chopin' fruit was characterized by lower, but still high in firmness. This cultivar stood out from the others, especially in 2019 and 2020; although in 2021, the value of the indicator in question for this cultivar fell within the same statistical group as clone 'JB' and 'Idared'. Clone 'JB', on the other hand, ranked together with 'Idared' and 'Ligol' in the same cultivar group. The Šampion cultivar differed from the other cultivars in that it stood out as having the lowest apple firmness during the three-year study. The value of the index fluctuated from one survey year to the next, and only in the case of 'Šampion' and clone 'JB' was no such relationship shown. The difference between the extreme values in the years depended on the cultivar and ranged from 9% ('Gala Brookfield', 'Ligol') to 12–13% ('Chopin', 'Idared').

**Table 2.** The values of firmness (N) for apples depending on cultivars and years of testing.

| Cultivars | Firmness | | | |
|---|---|---|---|---|
| | **2019** | **2020** | **2021** | ***p*-Value** |
| 'Gala Brookfield' | 85.5 ± 1.4 | 89.3 ± 0.9 | 81.9 ± 1.0 | <0.01 |
| 'Šampion' | 58.3 ± 0.4 | 58.7 ± 0.8 | 58.3 ± 0.1 | 0.75 |
| 'Ligol' | 65.8 ± 0.5 | 61.3 ± 0.5 | 60.3 ± 1.7 | <0.01 |
| clone 'JB' | 65.2 ± 3.6 | 68.6 ± 3.1 | 70.6 ± 1.6 | 0.25 |
| 'Chopin' | 70.1 ± 2.1 | 78.6 ± 0.8 | 70.3 ± 0.6 | <0.01 |
| 'Idared' | 65.0 ± 1.0 | 73.9 ± 1.3 | 72.5 ± 1.4 | <0.01 |
| *p*-value | <0.01 | <0.01 | <0.01 | |

Data are presented as mean ± standard deviation.

The analysis of the soluble solids content showed that this parameter depended on the cultivar used in the study as well as on the year the fruit was harvested. Apples in 2019 were characterized by a significantly higher SSC than in subsequent years (Table 3). The changes in SSC values observed in the experiment were not characterized by a unidirectional vector of the variable but were multidirectional depending on the cultivar. Thus, in 2019 'Idared', clone 'JB' and 'Ligol' can be placed in one group of cultivars with higher SSC in apple flesh. On the opposite side of the scale were 'Chopin' and 'Gala Brookfield' planned. In the following year 2020, again clone 'JB' had a high SSC, but this year 'Ligol', 'Chopin' and 'Idared' were placed in the lower SSC group. The year 2021 saw further changes and although 'Chopin' was a low SSC cultivar again, 'Gala Brookfield' and 'Šampion' joined this group again. On the other end of the scale, we had clone 'JB' for the third year in a row, as well as 'Idared' and 'Ligol'. Such a high variability in years between cultivars makes it possible to identify only 'Chopin' as the cultivar with low SSC and clone 'JB' with high SSC in apples. The high SSC in apples of clone 'JB' was determined by a high glucose content, almost three times higher than 'Chopin', 'Idared' or 'Ligol' (Table 4). Clone 'JB' was also characterized by high fructose content similar to the sweet cultivars, i.e., 'Gala Brookfield' and 'Šampion'. 'Chopin', on the other hand, was characterized by low levels of simple sugars. In the experiment, no significant differences were found in terms of sucrose content between the cultivars tested, regardless of the year of testing.

**Table 3.** The values of solids soluble content (°Brix) for apples depending on cultivars and years of testing.

| Cultivars | SSC | | | |
|---|---|---|---|---|
| | **2019** | **2020** | **2021** | ***p*-Value** |
| 'Gala Brookfield' | 13.0 ± 0.1 | 12.0 ± 0.1 | 11.6 ± 0.1 | <0.01 |
| 'Šampion' | 13.4 ± 0.2 | 12.4 ± 0.2 | 11.3 ± 0.1 | <0.01 |
| 'Ligol' | 13.8 ± 0.1 | 11.1 ± 0.1 | 12.5 ± 0.4 | <0.01 |
| clone 'JB' | 13.9 ± 0.1 | 12.9 ± 0.1 | 12.5 ± 0.2 | <0.01 |
| 'Chopin' | 12.9 ± 0.6 | 11.2 ± 0.1 | 11.5 ± 0.3 | <0.01 |
| 'Idared' | 14.2 ± 0.2 | 11.5 ± 0.1 | 12.4 ± 0.3 | <0.01 |
| *p*-value | <0.01 | <0.01 | <0.01 | |

Data are presented as mean ± standard deviation.

The statistical analysis of the titratable acidity of the cultivars studied distinguished two groups of apples characterized by extreme values of the index. The group of low-acid fruit included 'Ligol', 'Šampion' and 'Gala Brookfield', with the latter cultivar clearly indicated as having the lowest acid content in apple flesh (Table 5). The second group consisted of 'Idared', 'Chopin' and clone 'JB'. In this case, the apples of clone 'JB', but also

'Chopin', were characterized by very high titratable acidity (TA), exceeding 1% in almost every year of the study. The TA of the apples changed significantly depending on the year of the study. However, these changes did not cause significant differences between cultivars, as fruit from the first group contained from 0.28 to 0.65%, while the TA of apples from the second group ranged from 0.76 to 1.33% and was therefore on average two times higher. The high TA was influenced by the main hydroxy acids present in the apples, i.e., malic acid, citric acid and tartaric acid (Table 6). The content of the individual acids was determined by varietal characteristics. In general, malic acid is the main acid found in the fruit of the analysed cultivars, and the highest malic acid content was found in apples of clone 'JB', in which the value of this index was almost three times higher than in commonly grown apple cultivars. A high, but by more than 30% lower, content of malic acid was found in the cultivars 'Chopin' and 'Idared'. Clone 'JB' also contained more of the other identified acids, namely citric and tartaric. Unexpectedly, the 'Chopin' cultivar contained less citric acid than 'Gala Brookfield', and compared to 'Šampion', less tartaric acid.

**Table 4.** The values of sugar content (g·100 g$^{-1}$ FW) for apples depending on testing cultivars.

| Cultivars | Sugar | | |
|---|---|---|---|
| | Sucrose | Glucose | Fructose |
| 'Gala Brookfield' | 4.11 ± 0.31 | 0.76 ± 0.04 | 5.91 ± 0.32 |
| 'Šampion' | 3.68 ± 0.09 | 0.86 ± 0.02 | 5.86 ± 0.11 |
| 'Ligol' | 3.39 ± 0.87 | 0.34 ± 0.12 | 4.95 ± 1.41 |
| clone 'JB' | 5.39 ± 0.22 | 1.23 ± 0.15 | 5.55 ± 0.08 |
| 'Chopin' | 4.65 ± 1.59 | 0.40 ± 0.08 | 3.62 ± 0.15 |
| 'Idared' | 4.96 ± 0.15 | 0.43 ± 0.07 | 5.08 ± 0.10 |
| *p*-value | 0.14 | <0.01 | 0.02 |

Data are presented as mean ± standard deviation.

**Table 5.** The values of titratable acidity (%) for apples depending on cultivars and years of testing.

| Cultivars | Titratable Acidity | | | |
|---|---|---|---|---|
| | 2019 | 2020 | 2021 | *p*-Value |
| 'Gala Brookfield' | 0.28 ± 0.01 | 0.36 ± 0.01 | 0.47 ± 0.02 | <0.01 |
| 'Šampion' | 0.39 ± 0.02 | 0.58 ± 0.01 | 0.65 ± 0.01 | <0.01 |
| 'Ligol' | 0.42 ± 0.01 | 0.57 ± 0.01 | 0.64 ± 0.04 | <0.01 |
| clone 'JB' | 1.00 ± 0.03 | 1.33 ± 0.03 | 1.19 ± 0.02 | <0.01 |
| 'Chopin' | 0.90 ± 0.06 | 1.14 ± 0.01 | 1.08 ± 0.05 | <0.01 |
| 'Idared' | 0.76 ± 0.05 | 0.95 ± 0.01 | 0.88 ± 0.01 | <0.01 |
| *p*-value | <0.01 | <0.01 | <0.01 | |

Data are presented as mean ± standard deviation.

The content of total polyphenols in apples of the appraised cultivars was determined by varietal characteristics. Clone 'JB' was characterized by a two to three times higher TPC in the flesh than the other cultivars, although in 2021 'Idared' was also characterized by a similar TPC in apple flesh (Table 7). Assessing the total polyphenol content of the apple peel showed similar relationships between cultivars as seen in the flesh. The peel of clone 'JB' and 'Idared' contained two times more polyphenols than the same fruit element in the other cultivars. The evaluation of the apples made it possible to identify the cultivar whose fruit did not contain much total polyphenols, i.e., 'Ligol'. Regardless of the part of the fruit tested, this cultivar is characterized by a low content of compounds from the phenolic group. The content of total polyphenols in the flesh or skin of the other cultivars was

significantly higher than in 'Ligol' but also significantly lower than clone 'JB'. TPC content was also determined by weather conditions. A significant change in TPC depending on the year of the study was noted in the flesh of 'Šampion', 'Chopin', 'JB' clone and 'Idared', and in the peel of 'Šampion', 'Chopin' and 'Idared'. In general, it can be said that the listed cultivars were characterized by higher TPC in 2019 than in the subsequent years of the study.

**Table 6.** The values of acid content (g·100 g$^{-1}$ FW) for apples depending on testing cultivars.

| Cultivars | Acids | | |
|---|---|---|---|
| | **Malic** | **Citric** | **Tartaric** |
| 'Gala Brookfield' | 0.386 ± 0.019 | 0.069 ± 0.002 | 0.010 ± 0.002 |
| 'Šampion' | 0.541 ± 0.010 | 0.046 ± 0.001 | 0.012 ± 0.001 |
| 'Ligol' | 0.413 ± 0.111 | 0.031 ± 0.009 | 0.007 ± 0.002 |
| clone 'JB' | 1.130 ± 0.003 | 0.081 ± 0.004 | 0.014 ± 0.003 |
| 'Chopin' | 0.778 ± 0.053 | 0.041 ± 0.001 | 0.011 ± 0.003 |
| 'Idared' | 0.656 ± 0.010 | 0.025 ± 0.007 | 0.006 ± 0.001 |
| *p*-value | <0.01 | <0.01 | 0.01 |

Data are presented as mean ± standard deviation.

**Table 7.** The values of total polyphenols content (mg·100 g$^{-1}$ FW) for flesh and peel of apples depending on cultivars and years of testing.

| Cultivars | Years | | | *p*-Value |
|---|---|---|---|---|
| | **2019** | **2020** | **2021** | |
| | | Flesh | | |
| 'Gala Brookfield' | 158 ± 13 | 142 ± 5 | 170 ± 24 | 0.28 |
| 'Šampion' | 228 ± 2 | 150 ± 1 | 163 ± 8 | <0.01 |
| 'Ligol' | 133 ± 18 | 93.0 ± 5 | 98.3 ± 15 | 0.06 |
| clone 'JB' | 531 ± 10 | 504 ± 6 | 211 ± 34 | <0.01 |
| 'Chopin' | 212 ± 24 | 102 ± 5 | 126 ± 27 | <0.01 |
| 'Idared' | 279 ± 22 | 107 ± 7 | 205 ± 12 | <0.01 |
| *p*-value | <0.01 | <0.01 | <0.01 | |
| | | Peel | | |
| 'Gala Brookfield' | 782 ± 25 | 802 ± 50 | 740 ± 28 | 0.29 |
| 'Šampion' | 743 ± 51 | 777 ± 19 | 453 ± 29 | <0.01 |
| 'Ligol' | 368 ± 23 | 300 ± 35 | 288 ± 48 | 0.14 |
| clone 'JB' | 1289 ± 33 | 1273 ± 43 | 1282 ± 20 | 0.90 |
| 'Chopin' | 632 ± 19 | 623 ± 6 | 509 ± 10 | <0.01 |
| 'Idared' | 1236 ± 44 | 1276 ± 20 | 935 ± 98 | <0.01 |
| *p*-value | <0.01 | <0.01 | <0.01 | |

Data are presented as mean ± standard deviation; FW—fresh weigh.

Among the apple cultivars assessed, 'Idared' stood out for its high content of phenolic acids in the flesh (Table 8). In all years of the study, the indicated cultivar was characterized by a two-fold higher content of phenolic acids in the flesh than cultivars with a low content of these compounds. The group of cultivars with high phenolic acid content also included clone 'JB' (years 2019 and 2020) and 'Gala Brookfield' (in 2020). On the other end of the scale were cultivars with low levels of phenolic acids in the apple flesh and we can include 'Šampion' in 2019 and 2021 and 'Ligol' and 'Chopin' in 2020 in this group. The level of phenolic acids in the peel of the apples evaluated was variable depending on the genetic characteristics of the cultivar and the year of the study. However, the year of the study determined changes in the phenolic acid content of apple peel only in cultivars with a low

phenolic acid content. During the three-year study, it was found that the cultivars 'Idared', 'Gala Brookfield' and clone 'JB' were characterized by a high and constant level of phenolic acids in the peel. Low levels of phenolic acids were found in the peel of 'Šampion' apples in 2019 and 2021, in 2020 and 2021 in 'Ligol' and in 2019 in 'Chopin'.

**Table 8.** The values of phenolic acids content (mg·100 g$^{-1}$ FW) for flesh and peel of apples depending on cultivars and years of testing.

| Cultivars | Years | | | *p*-Value |
|---|---|---|---|---|
| | **2019** | **2020** | **2021** | |
| | Flesh | | | |
| 'Gala Brookfield' | 8.7 ± 0.1 | 13.9 ± 0.7 | 8.5 ± 0.2 | <0.01 |
| 'Šampion' | 4.7 ± 0.1 | 11.7 ± 0.1 | 7.0 ± 0.2 | <0.01 |
| 'Ligol' | 7.2 ± 0.5 | 8.8 ± 0.2 | 8.8 ± 0.4 | <0.01 |
| clone 'JB' | 15.2 ± 0.1 | 15.3 ± 0.1 | 9.3 ± 0.3 | <0.01 |
| 'Chopin' | 10.1 ± 0.4 | 9.4 ± 0.5 | 9.0 ± 0.2 | 0.06 |
| 'Idared' | 14.5 ± 1.6 | 14.3 ± 0.3 | 15.4 ± 0.1 | 0.48 |
| *p*-value | <0.01 | <0.01 | <0.01 | |
| | Peel | | | |
| 'Gala Brookfield' | 45.7 ± 7.3 | 55.5 ± 5.0 | 41.9 ± 0.4 | 0.09 |
| 'Šampion' | 28.9 ± 0.5 | 51.2 ± 1.2 | 24.2 ± 3.3 | <0.01 |
| 'Ligol' | 31.6 ± 0.6 | 17.3 ± 0.1 | 23.5 ± 3.4 | <0.01 |
| clone 'JB' | 46.0 ± 1.2 | 48.7 ± 1.0 | 46.5 ± 0.5 | 0.06 |
| 'Chopin' | 24.9 ± 0.7 | 39.1 ± 0.6 | 32.6 ± 2.3 | <0.01 |
| 'Idared' | 51.6 ± 3.6 | 53.5 ± 0.9 | 49.6 ± 0.9 | 0.28 |
| *p*-value | <0.01 | <0.01 | <0.01 | |

Data are presented as mean ± standard deviation; FW—fresh weigh.

During a three-year study, it was found that the flavonols content of apple flesh was only valued by varietal traits in one year. It was then shown that cultivars containing more polyphenols or phenolic acids ranked lower in terms of flavonols content (Table 9). However, this thesis was not proven in subsequent years of the study, and the value of the index was highly variable between years, making it impossible to show any trends. The flavonols content was influenced by the year of the study. It turned out that the studied cultivars, with the exception of 'Gala Brookfield', had a higher content of flavonols in the flesh in 2021 than in 2019 or 2020. In contrast, the trends describing the flavonols content of apple peel are similar to the observed relationships for total polyphenols or phenolic acids. In this case, apples of clone 'JB' were again characterized by high levels of flavonols in the peel, while 'Šampion' and 'Ligol' apples were characterized by low levels of these compounds. Unexpectedly, 'Idared' was also characterized by a low flavonols content in the apple peel. Assessing the impact of the year of the study, it was found that apples in 2020 had a higher flavonols content in the peel than in the other two years of the study.

The content of antioxidant compounds determines the antioxidant activity of the fruit. Clone 'JB' was characterized by high AA in apple flesh in all years of the study (Table 10). In the subsequent years of the study, the of clone 'JB' flesh was 107%, 70% and 178% higher than that of 'Ligol', the cultivar characterized by the lowest value of this parameter, respectively. The cultivar 'Ligol' also stood out in terms of the level of AA of the apple peel. In this evaluation, AA in the 'Ligol' peel was again found to be lower than in the other cultivars. In contrast, the differences between the other cultivars assessed were not so significant and only in 2021 'Šampion' and 'Gala Brookfield' apples had lower levels of AA than in the skins of clone 'JB', 'Idared' or 'Chopin' apples. AA variability between study years was observed for 'Šampion' and 'JB' clone (skin and flesh), 'Gala Brookfield' only in peel and 'Chopin' only in flesh. It should be noted that in 2020 'Šampion' had the highest AA value, while 'Chopin' and 'JB' clone had the lowest AA value in fruit flesh. On

the other hand, the AA of the peel of the 'Gala Brookfield', 'Šampion' and 'JB' clone was higher in 2019 than in 2020.

**Table 9.** The values of flavonols (mg·100 g$^{-1}$ FW) for the flesh and peel of apples depending on cultivars and years of testing.

| Cultivars | Years | | | *p*-Value |
|---|---|---|---|---|
| | **2019** | **2020** | **2021** | |
| | Flesh | | | |
| 'Gala Brookfield' | 134 ± 17 | 116 ± 16 | 136 ± 2 | 0.28 |
| 'Šampion' | 107 ± 7 | 118 ± 16 | 141 ± 2 | 0.04 |
| 'Ligol' | 119 ± 14 | 104 ± 2 | 143 ± 1 | 0.01 |
| clone 'JB' | 108 ± 2 | 130 ± 15 | 141 ± 1 | 0.02 |
| 'Chopin' | 107 ± 1 | 115 ± 13 | 139 ± 1 | 0.01 |
| 'Idared' | 105 ± 1 | 92.6 ± 19 | 141 ± 1 | 0.01 |
| *p*-value | 0.03 | 0.24 | 0.38 | |
| | Peel | | | |
| 'Gala Brookfield' | 312 ± 4 | 317 ± 23 | 311 ± 1 | 0.91 |
| 'Šampion' | 212 ± 2 | 326 ± 23 | 301 ± 10 | <0.01 |
| 'Ligol' | 238 ± 32 | 288 ± 33 | 344 ± 39 | 0.06 |
| clone 'JB' | 263 ± 1 | 407 ± 4 | 362 ± 5 | <0.01 |
| 'Chopin' | 268 ± 30 | 319 ± 29 | 297 ± 12 | 0.21 |
| 'Idared' | 226 ± 2 | 315 ± 30 | 288 ± 16 | 0.01 |
| *p*-value | <0.01 | <0.01 | 0.01 | |

Data are presented as mean ± standard deviation; FW—fresh weigh.

**Table 10.** The values of antioxidant activity (mg AAE·100 g$^{-1}$ FW) for flesh and peel of apples depending on cultivars and years of testing.

| Cultivars | Years | | | *p*-Value |
|---|---|---|---|---|
| | **2019** | **2020** | **2021** | |
| | Flesh | | | |
| 'Gala Brookfield' | 0.32 ± 0.02 | 0.32 ± 0.02 | 0.40 ± 0.03 | 0.02 |
| 'Šampion' | 0.28 ± 0.02 | 0.38 ± 0.01 | 0.33 ± 0.01 | <0.01 |
| 'Ligol' | 0.25 ± 0.03 | 0.27 ± 0.02 | 0.19 ± 0.01 | 0.02 |
| clone 'JB' | 0.58 ± 0.02 | 0.46 ± 0.01 | 0.53 ± 0.03 | <0.01 |
| 'Chopin' | 0.33 ± 0.03 | 0.25 ± 0.01 | 0.37 ± 0.02 | <0.01 |
| 'Idared' | 0.29 ± 0.03 | 0.36 ± 0.01 | 0.35 ± 0.02 | 0.03 |
| *p*-value | <0.01 | <0.01 | <0.01 | |
| | Peel | | | |
| 'Gala Brookfield' | 0.74 ± 0.01 | 0.72 ± 0.01 | 0.68 ± 0.01 | <0.01 |
| 'Šampion' | 0.74 ± 0.01 | 0.74 ± 0.01 | 0.66 ± 0.03 | <0.01 |
| 'Ligol' | 0.65 ± 0.02 | 0.62 ± 0.01 | 0.60 ± 0.01 | 0.01 |
| clone 'JB' | 0.73 ± 0.01 | 0.69 ± 0.01 | 0.74 ± 0.02 | <0.01 |
| 'Chopin' | 0.71 ± 0.02 | 0.71 ± 0.01 | 0.70 ± 0.01 | 0.46 |
| 'Idared' | 0.74 ± 0.01 | 0.70 ± 0.01 | 0.71 ± 0.03 | 0.55 |
| *p*-value | <0.01 | <0.01 | <0.01 | |

Data are presented as mean ± standard deviation; FW—fresh weigh.

## 4. Discussion

In recent years, 'healthy lifestyles' based on sports participation and appropriate nutrition programmes have become increasingly important. New diets are being developed with the aim of preventing weight loss and civilization diseases, such as diabetes and cardiovascular disease. The foundation on which most nutrition programmes are based is

fruit and vegetables, and the prevalence and availability of apples makes them an ideal component of diets [38,39]. In general, more and more attention is being paid to products that can be categorized as so-called functional foods, i.e., products with above-average nutritional properties. Apples, and in particular certain cultivars characterized by such properties, can be included in this group and become a staple ingredient in diets [40]. Consumers' choice of apple is not determined solely by its nutritional properties but by its appearance and taste [26,27,41,42]. Apples contain monosaccharides; however, these are easily digested and rapidly utilized by the human body. In studies, the most abundant sugar in apples of cultivated cultivars was fructose followed by sucrose [43–45]. Our own research showed significant differences between commonly grown apple cultivars in Poland and 'Chopin' or 'JB' clone. Apple cultivars considered sweet by consumers, such as 'Šampion' and 'Gala', were characterized by lower sugar and SSC but also low TA. The high SSC is largely due to monosaccharides, the levels of which were high in apples of clone 'JB'. Many studies raise the issue of the determination of sugars, acids by varietal characteristics [41,43] and the content of individual sugars, which can vary considerably within apple cultivars. According to Li et a. [44], wild apple cultivars produce fruit that is more acidic than cultivated cultivars, which is strongly influenced by genetic background and growing location. In the experiment, we found that 'Chopin' apples and clone 'JB' apples had above-average organic acid content, regardless of the year of the study. Despite their sour taste, the apples deacidify the body because they are rich in the alkaline potassium (regulates water balance) and iron (prevents anaemia). In our study, it was noted that the soluble solids content and titratable acidity of apples are influenced by weather conditions and especially important is the period preceding fruit harvest. In 2019 (a dry and warm year), all evaluated cultivars were characterized by higher soluble solids content than in subsequent years. However, in 2020 and 2021, cultivars harvested in late August and September ('Gala Brookfield', 'Šampion') were characterized by a different SSC and TA than those harvested more than a month later, which was probably related to the frequency of precipitation occurring in both years. The data obtained in the experiment confirm previous reports on the significance of the effect of weather conditions on apple quality [45–47].

As with other types of fruit, most of the valuable substances in apples are found in the peel and just below the peel [21,48], but unfortunately many consumers eat apples after they have been peeled. Numerous studies show that the peel of apples is much richer in phenolic compounds than the flesh [21,49,50]. The polyphenol content of apples is influenced by many factors, such as the varietal factor [21,50] or the degree of ripeness [18]. Koutsos et al. [51] demonstrated that the consumption of apples or their bioactive components is associated with beneficial effects on lipid metabolism and other markers of cardiovascular disease. The effects of flavanol rich foods, such as apples on blood pressure, may be due to the monomeric flavanols, mainly (-)-epicatechin, and the oligomeric flavanols, i.e., procyanidins [52,53]. Our own research indicates that there were significant differences in the composition of the bioactive compounds of the cultivars evaluated. Clone 'JB' was particularly rich in compounds from the polyphenol group in both the skin and flesh of apples. 'Idared' also deserves a distinction in terms of total polyphenol content. On the other hand, the fruit of the 'Chopin' cultivar had an average content of total polyphenols and other antioxidants, i.e., phenolic acids and flavonols. Distinctive clone 'JB' was in the forefront among cultivars with a high content of antioxidant compounds, although the level of individual antioxidants was determined by the year of study. Differences in the content of biologically active compounds may be due to both differences between cultivars and the maturity of the fruit analysed, as pointed out by MacLean et al. [54]. In a study [55,56], polyphenol content and antioxidant activity were found to differ significantly between the cultivars studied, with higher polyphenol content, accompanied by excellent antioxidant activity both in the flesh and in the whole fruit.

The nutritional properties of apples largely depend on the cultivar [57,58], the type of cultivation [59] weather conditions [60,61] and agrotechnical aspects [62]. As for envi-

ronmental factors, the role of weather conditions, such as temperature and precipitation, is crucial and has only been discussed in a limited number of research papers. Warrington et al. [63] found that fruit grown at higher temperatures had higher average weight, higher soluble solids content and lower starch concentration at harvest maturity. Regarding polyphenol formation and antioxidant capacity in fruit, studies suggest that heat stress may be a trigger, and that low temperatures may stimulate the expression of genes involved in, for example, anthocyanin synthesis in apple peel [64,65]. Additionally, among polyphenols, flavonols are the most weather-dependent; the degree of exposure to sunlight is suggested as an underlying mechanism [66]. According to Zang et al. [67] average, minimum and maximum temperatures from April to October are the main influences on fruit quality. The influence of climatic conditions was the significant factor determining the nutritional properties of apples in the study. In the warm and rainless year of 2019, most cultivars showed higher TPC than in subsequent years. However, the content of individual compounds was variable and dependent on varietal characteristics and climatic conditions. The experiment confirmed that temperatures and precipitation occurring in the period before fruit ripening affect the content of biologically active compounds in apples [68–70]. High temperature during apple ripening, can lead to the increased activity of antioxidant enzymes. In response, the plant produces an antioxidant defence system composed of enzymes or increased amounts of antioxidants [45,56]. Many of these processes can be supported by photosynthetic activity [71]. Therefore, it is very important to select apple cultivars with the highest content of phenolic compounds and to determine exactly how the different factors affect phenolic levels.

## 5. Conclusions

The present study showed diversity in the cultivars, focusing on the quality of fruit as well as their nutritional value. In a three-year study, the average fruit acidity of the 'Chopin' and 'JB' clone was found to be almost three times higher than in fruits commonly grown in Poland. An important feature of cultivars with red flesh, such as the JB clone, is of significantly higher content of antioxidant compounds than in other cultivars. The content of antioxidant compounds determines the antioxidant activity of the fruit, so the flesh of the 'JB' clone apples was characterized by high antioxidant activity. However, weather conditions strongly affect the physicochemical quality and content of biologically active compounds in apples. In dry and warm summers, fruits may contain more monosaccharides and less organic acids. In addition, their antioxidant profile is modified. Higher air temperatures and a lack of precipitation in the period before apples ripen can contribute as stress factors to an increase in the polyphenol content of apple flesh and peel.

**Supplementary Materials:** The following supporting information can be downloaded at: https://www.mdpi.com/article/10.3390/agriculture12111876/s1, Figure S1: Average monthly temperatures and monthly precipitation in 2019; Figure S2: Average monthly temperatures and monthly precipitation in 2020; Figure S3: Average monthly temperatures and monthly precipitation in 2021.

**Author Contributions:** Conceptualization, A.K., D.W. and T.K.; methodology, A.K., D.W. and T.K.; formal analysis, A.K., D.W. and T.K.; investigation, A.K., D.W. and T.K.; writing—original draft preparation, A.K., D.W. and T.K., writing—review and editing, A.K., D.W. and T.K. All authors have read and agreed to the published version of the manuscript.

**Funding:** This research received no external funding.

**Institutional Review Board Statement:** Not applicable.

**Data Availability Statement:** Not applicable.

**Acknowledgments:** The authors would like to give thanks to A. Przybyla and E. Pitera for providing the fruits for the research.

**Conflicts of Interest:** The authors declare no conflict of interest.

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
