# Peer review of "Quality and Nutritional Value of ‘Chopin’ and Clone ‘JB’ in Relation to Popular Apples Growing in Poland"

_agriculture, doi:10.3390/agriculture12111876_

Round 1
Reviewer 1 Report
Comments to the Manuscript ID: agriculture-1973908
1. The Abstract needs editing. Some sentences are repeated by semantic meaning. The Abstract must contain a generalized conclusion of the study, and not a statement of facts.
2. “Introduction” section: this part should be shortened, formulated briefly, since this part of the manuscript contains well-known information. The title of the manuscript indicates a sectional program, but nothing is presented in the text about it. It is necessary to include information about the SGGW breeding program, i.e. the purpose of the program, target indicators, developers and participants of the program. Indicate similar breeding programs that have been developed in the world and what general trends there are.
3. How will the SGGW breeding program contribute to the development of your country's economy, namely the development of the horticultural industry?
4. “Methods and Materials” section:
Point out the genetic origin of the ‘Chopin’ cultivar and ‘JB’ clone. Give brief description of ‘Chopin’ and ‘JB’, their advantages and disadvantages. Line 96-97: The information is not clear: did the authors determine the analysis of the physical and chemical parameters of the fruits in frozen samples? In this case, was it an additional factor of the experiment, or how did the authors exclude it?
Line 118-119: why was the titrated acidity evaluated in diluted juice? Are these research results reliable?
Statistical analysis: more detailed information should be added to this section.
5. “Results” section: What explains the increase in 2020 and the decrease in 2021 in the firmness of fruits of some apple cultivars?. If this is due to meteorological factors, why don't the authors indicate this? Line 243-246: is the data presented here after freezing? I don't understand.
6. Were there significant differences between the new genotypes of Chopin’and ‘JB’ apples and the studied well-known cultivars of apples by physical and chemical parameters? What main chemical properties of ‘Chopin’ and ‘JB’ apples will allow them to be used for a healthy lifestyle?
7. “Discussion” section: The section needs to be finalized.
Line 284-304: the information on the semantic meaning is repeated from the "Introduction" section. ‘Gala Brookfield’, ‘Idared’, ‘Šampion’ and ‘Ligol’ are well-known industrial apple cultivars; the biochemical fruit evaluation of these cultivars has been studied in different geographical areas, so provide the information about the variability of the physical and chemical properties of apples and compare with your obtained results.
8. The section "Conclusions" does not correspond to the stated aim of the research by the authors. This section needs to be finalized.
Author Response
Thank you for your detailed and relevant comments on the manuscript. Please see the attachment.

Reviewer 2 Report
Dear authors,
Please, find attached my comments within the pdf file

Author Response

(The authors gave the same response as above.)

Author Response

(The authors gave the same response as above.)

Reviewer 4 Report
This manuscript showed the contents of various nutrients in six apple varieties and evaluated the differences between varieties and years.
There seem to be two main problems.
1. The contents of nutritional components in fruits could be vary greatly depending on the degree of crop load of the trees sampled. It also varies depending on the size of the fruit, the degree of maturity, and weather factors. It is not possible to determine whether the contents are genetically different depending on the cultivar since the degree of crop load in the trees from which the fruit were sampled has not been shown. Moreover, even if the content varies from year to year, it is not possible to interpret the reason for this because the weather conditions and fruit maturity of each year are not shown. It seems meaningless to simply compare the measured values without considering the environmental factors that affect the contents.
2. It is natural that the contents of nutritional components differ depending on the variety. It is important whether the difference in the contents is meaningful or not. It is not possible to judge whether the difference in the contents between varieties is meaningful in terms of the value of diet food. For example, if the content of a nutrient component is much lower than the daily intake that is effective for health, it can be said that eating any kind of fruit will have little effect on health. If so, the significant difference in content between varieties is meaningless from the point of view of diet foods. It is necessary to consider whether this is really the case.
Author Response

(The authors gave the same response as above.)

Author Response

(The authors gave the same response as above.)

Round 2
Reviewer 1 Report
Thanks to the authors for the answers to my questions.
The authors have significantly improved the article, taking into account my comments.
However, I suggest deleting lines 302-307, because it contains known information.
To add some information (improve) to the "Discussion" section about the influence of climatic factors of different regions on the change in biochemical quality indicators of apple varieties 'Gala Brookfield', 'Idared', 'Šampion' and 'Ligol'.
Reviewer 2 Report
No comments
Author Response
Dear Reviewer,
Thank you for your comments and positive acceptance of the article for publication in Agriculture
Reviewer 3 Report
The authors have addressed all the specific issues I raised in my initial review in a satisfactory way.
Author Response

(The authors gave the same response as above.)

Reviewer 4 Report
I think it's revised properly.
Author Response

(The authors gave the same response as above.)

Reviewer 5 Report
The Authors corrected and revised all issue from previous Manuscript and I accept this revised Manuscript for publishing.
Author Response

(The authors gave the same response as above.)
